# Development and Evaluation of Loop-Mediated Isothermal Amplification (LAMP) as a Preliminary Diagnostic Tool for Brown Root Rot Disease Caused by *Phellinus noxius* (Corner) G. H. Cunningham in Hong Kong Urban Tree Management

**Hao Zhang** [1,*] **, Tze Kwun Ng** [2] **, Kai Chun Lee** [1] **, Zoen Wing Leung** [1] **, Wai Fu Yau** [2] **and Wai Shing Wong** [2]

1 Faculty of Design and Environment, Technological and Higher Education Institute of Hong Kong, Hong Kong, China
2 Landscape Division, Highways Department, HKSAR Government, Hong Kong, China
* Correspondence: allenzh@vtc.edu.hk; Tel.: +852-3890-8282

**Abstract:** Brown Root Rot Disease (BRRD) is one of the most devastating urban tree diseases in tropical and subtropical areas, including Hong Kong. It can result in tree death in a few months and is difficult to detect in the early stages of development. Fungal isolation and PCR methods are currently the most widely adopted methods to diagnose the disease. However, they are both time and technically demanding. Loop-mediated isothermal amplification (LAMP) is a superior molecular-based diagnostic method with great specificity, accessibility, and effectivity. In this study, 15 BRRD-positive and 15 BRRD-negative trees were sampled from 19 roadside slopes in Hong Kong from the end of 2020 to the middle of 2021. The wood tissues were isolated and cultivated in PN3 and PDA agars for the disease diagnosis. The mycelium samples in PDA were directly conducted in LAMP kits (mLAMP) to substitute the purified DNA materials. Wood tissues were also used in LAMP kits (wLAMP) as impurified and highly contaminated samples. The results of mLAMP and wLAMP were compared with the results of isolation to evaluate the specificity and sensitivity of LAMP method. The results showed that mLAMP had 100% sensitivity and 73.3% specificity. For wLAMP, both the sensitivity and specificity were 73.3%. For symptomless trees, 85.7% and 64.3% congruencies were found in mLAMP and wLAMP, respectively. Based on the results of this study, the co-application of LAMP in the current tree management work was also discussed. We envisaged LAMP is a sensitive, prompt, and user-friendly method to diagnose BRRD and it could favor the BRRD diagnosis in fields by accelerating and promoting large-scale screening.

**Keywords:** loop-mediated isothermal amplification; fungal isolation; direct LAMP; Brown Root Rot Disease (BRRD); *Phellinus noxius*

## 1. Introduction

Brown Root Rot Disease (BRRD) is currently reported as one of the most insurmountable problems in urban forest management of many tropical and subtropical regions due to its highly infectious nature and devastating impacts. *Phellinus noxius* (Corner) G. H. Cunningham is a white rot fungus that can induce BRRD. Taxonomically, *P. noxius* is a fungus in the family Hymenochaetaceae, order Hymenochaetales, phylum Basidiomycota of the kingdom Fungi [1]. Similar to the vast majority of species in the Hymenochaetaceae family, *P. noxius* grows mycelium and rigid basidiocarps for reproduction. Trees infected with BRRD may ultimately die or fall due to root failure, as the infected root systems fail to transport nutrients or provide good structural support to trees [2,3].

Urban trees provide a broad range of functions to a city, such as purifying the air [4], alleviating the heat island effect [5], promoting urban biodiversity [6], and offering an exquisite nature scenery to citizens [7]. However, a failure to identify hazard trees, such as

those with BRRD, poses a great risk to citizens. In particular, early detection of BRRD is vital as it can cause serious and irreversible damages to trees. In some cases, tree death may occur within months after the start of root rotting [2]. Furthermore, *P. noxius* has a wide host range (e.g., *Acacia confusa* Merr., *Ficus microcarpa* L. f. and *Prunus persica* (L.) Batsch) in tropical and subtropical regions and can disperse via soils and root segments [8]. The highly infectious nature of the disease can result in a widespread occurrence of the disease.

Visual assessment at field is one way of identifying BRRD. Hyphae of *P. noxius* invade and degrade lignin and cellulose of root cells. The mycelium grows into the inner roots and aggregates into yellow-brown mycelial nets. Fruiting bodies may be sometimes present [3]. The symptoms of BRRD include discolored wood, die-back twigs, and abnormal leaf condition, which are due to the impaired roots. [2]. However, while visual assessment may allow a quick diagnosis of BRRD, this method has great limitations. Obvious symptoms are usually visible at the later stage of infection. In addition, many symptoms are not specific to BRRD.

Early detection and accurate diagnosis of BRRD can be achieved through laboratory diagnosis. Previously, agar isolation was widely adopted to diagnose BRRD. The identification was based on the presence of the colony of *P. noxius*, which are irregular dark brown patches or lines [2]. With the development of biotechnology, molecular identification has become an alternative popular tool for diagnosis. The DNA sequencing method relying on a polymerase chain reaction (PCR) is the prevalent identification method [9]. It allows the analysis of diseases at a species-specific level and establishment of a gene map based on the found disease genome [8]. However, it is time-consuming and technically demanding.

In recent years, Loop-mediated Isothermal Amplification (LAMP) is a molecular diagnostic tool being increasingly used in different disciplines as an alternative for PCR. It provides simple but critical preliminary binary diagnosis results ("positive" or "negative") for informing further actions. Briefly, LAMP primers first attach to the target double-stranded DNA (dsDNA). Following that, DNA polymerase replicates the DNA segments into a single-stranded DNA (ssDNA). By applying the specifically designed primers, the end of the ssDNA can form a self-hybridizing loop and a dumbbell structure. In other words, the denaturation step will be eliminated, which is the critical step in separating the dsDNA in the PCR method [10]. It is time-saving and supports the operation of LAMP at a steady temperature. In addition, as more primers (4–6) are applied in the LAMP, the sensitivity of the test is higher compared to the PCR method [11,12]. Moreover, LAMP is reported to have greater tolerance of undesirable sample purity [9], curtailing the stringent DNA extraction process [10].

In Hong Kong, the first case of BRRD was reported in 2008 [13]. In recent years, the number of BRRD infected trees has increased exponentially [8]. There is a need to establish a rapid and effective diagnosis method for the disease and incorporate it as a part of the tree risk management strategy. LAMP is a potential tool. Currently, most studies that apply LAMP are on human, animal, and crop pathogens [11,14,15]. Studies focusing on the application of LAMP on urban trees for disease diagnosis are still scarce. Therefore, this study aimed to evaluate the accuracy and sensitivity of LAMP method in the diagnosis of BRRD by using two different types of sample materials, mycelium, and wood chips. In addition, the co-application of the LAMP methods with the current tree management in Hong Kong was elevated to provide a comprehensive discussion, as well as explore the potential application of LAMP in urban forest management.

## 2. Materials and Methods

### 2.1. Study Area

The study was carried out in Hong Kong, China (latitude 22° N and longitude 114° E) from 2020 to 2021. Hong Kong has a humid subtropical climate (Köppen climate classification: Cwa), which favors the development and dissemination of BRRD [8]. The monthly average temperatures of the coldest and hottest months are 16.3 °C (January) and 28.8 °C (July), respectively, and relative humidity exceeds 70% in all months except December [16].

Hong Kong is a metropolis with a severe shortage of flatland and with abundant rugged terrain ranging to 957 m [17]. The unique landscape imprints the assemblage of vegetative slopes in human dense areas. The studied trees were distributed on 18 different roadside slopes managed by the Highways Department of HKSAR Government.

### 2.2. Sample Collection

Samples were collected from a total of 30 trees, with 15 trees with BRRD (BRRD-positive trees) and 15 trees free of BRRD (BRRD-negative trees), from the end of 2020 to the middle of 2021 (Table 1). To avoid contamination, all tools used for sample collection were sterilized with 75% alcohol. Living root tissues were collected below the root collars, where *P. noxius* is likely to grow [18]. Four samples were collected in four directions (East, South, West, and North) of each tree and stored in zipper bags. All samples were preserved in refrigerators at a temperature of 2 °C before conducting experiments in the laboratories.

**Table 1.** Species, location, BRRD symptoms, of the 15 BRRD-positive trees and 15 BRRD-negative trees.

| Species [1] | Location (Slope No.) [2] | BRRD Symptoms | BRRD | Code |
|---|---|---|---|---|
| *Acacia confusa* Merr. | 11SW-A/F31 | Absence | Yes | P01 |
| | 11SW-D/CR1168 | Sparse foliage density; suspected root rotting | No | N01 |
| *Aleurites moluccanus* (L.) Willd. | 11SW-A/CR185 | Absence | Yes | P02 |
| *Cascabela thevetia* (L.) Lippold | 11SW-A/CR80 | Absence | No | N02 |
| | 11SW-C/FR43 | Sparse foliage density; medium density of die-back twigs | No | N03 |
| *Celtis sinensis* Pers. | 3 SW- C/F 35 | Absence | No | N04 |
| *Cinnamomum camphora* (L.) J. Presl | 11SE-A/C324 | Suspected root rotting | Yes | P03 |
| *Ficus microcarpa* L. f. | 11SW-A/CR80 | Abnormal bark color | No | N05 |
| | 11SW-A/R1148 | Absence | No | N06 |
| | 11SW-A/R427 | Abnormal bark color | No | N07 |
| | 3 SW- C/F 35 | Sparse foliage density; medium density of die-back twigs | No | N08 |
| *Ficus variegata* Blume | 11SE-A/C211 | Absence | No | N09 |
| | 11SE-A/C8 | Absence | No | N10 |
| | 11SW-A/CR185 | Absence | Yes | P04 |
| | 11SW-A/F31 | Absence | Yes | P05 |
| | 11SW-C/C341 | Absence | Yes | P06 |
| *Litsea glutinosa* (Lour.) C. B. Rob. | 11SW-D/C385 | Sparse foliage density; medium density of die-back twigs; abnormal bark color | No | N11 |
| *Livistona chinensis* (Jacq.) R. Br. ex Mart. | 11SW-D/CR239 | Sparse foliage density; suspected root rotting; observable mycelial nets | Yes | P07 |
| | 11SW-D/CR239 | Sparse foliage density; suspected root rotting; observable mycelial nets | Yes | P08 |
| | 11SW-D/CR239 | Sparse foliage density | Yes | P09 |
| *Lophostemon confertus* (R. Br.) Peter G. Wilson & J. T. Waterh. | 11SE-A/CR565 | High density of die-back twigs; suspected root rotting | Yes | P10 |
| | 11SE-A/CR565 | Absence | Yes | P11 |

**Table 1.** *Cont.*

| Species [1] | Location (Slope No.) [2] | BRRD Symptoms | BRRD | Code |
|---|---|---|---|---|
| | 11SW-A/FR258 | Medium density of die-back twigs | Yes | P12 |
| *Macaranga tanarius* var. *tomentosa* (Blume) Müll. Arg. | 11SW-C/C387 | Sparse foliage density; abnormal leaf size; low density of die-back twigs; abnormal bark color | No | N12 |
| | 11SW-C/F396 | Absence | No | N13 |
| | 11SW-C/FR43 | Absence | No | N14 |
| *Machilus chekiangensis* S. K. Lee | 11SW-D/CR239 | Abnormal bark color | No | N15 |
| | 11SW-D/CR239 | Abnormal bark color | Yes | P13 |
| *Scolopia saeva* Hance | 11SW-D/C385 | Medium density of die-back twigs; abnormal bark color | Yes | P14 |
| *Senna siamea* (Lam.) H. S. Irwin & Barneby | 11SW-C/FR43 | Absence | Yes | P15 |

[1] Species names are reported on: https://powo.science.kew.org/ accessed on 25 May 2022. [2] Details of slope: https://hkss.cedd.gov.hk/hkss/en/facts-and-figures/slope-information-system/sis/index.html/ accessed on 25 May 2022.

### 2.3. Experimental Design and Setup

The experiments were carried out in the laboratories of the Technological and Higher Education Institute of Hong Kong (THEi), Chai Wan, Hong Kong, from the end of 2020 to the middle of 2021.

### 2.3.1. Fungal Isolation

To prepare samples for isolation, the wood tissue sample was first chopped into a few 3 mm cubes and cleaned using deionized water to eliminate any undesired microorganisms. The samples were treated with sodium hypochlorite for 30 s for sterilization. The sodium hypochlorite was filtered, and the samples were further immersed in distilled water to remove attached sodium hypochlorite. The samples were collected and dried on tissue paper. After drying, 4–6 pieces of wood chip were inoculated in PN3 agars (20 g malt-extract, 20 g nutrient agar, 30 mg benomyl, 10 mg dicloran, 100 mg ampicillin, and 500 mg gallic acid per liter) [19–21], which can eliminate repulsive soil bacteria or fungi. The inoculated samples were stored in an incubator at 25–30 °C, an optimum temperature range for *P. noxius* growth. The samples were preserved in the incubator for 4 days for mycelium growing [22]. Morphological information including colony colors, textures, forms, and other cultural appearances was recorded. Samples with observable white or yellow-brown colors and irregular shape mycelium indicated the potential presence of *P. noxius* [23]. Part of the mycelium tissues was transferred into potato dextrose agar (PDA), which provided better nutrient resources for fungal growth [24]. After inoculating for five to six days, samples with observable brown color and irregular shape mycelium (Figure 1A) were processed under a microscope for micro-identification. BRRD was confirmed if trichocysts, a complex and specific form of mycelium of *P. noxius* [2], were observed (Figure 1B).

### 2.3.2. LAMP Method Using Mycelium (mLAMP)

The 3 mm cube samples of root tissues were ground into powder using a mortar. Liquid nitrogen was added before grinding to release DNA. The powders were then dissolved in 600 μL of deionized water. Then, 400 μL of the solution was centrifuged at room temperature at 14,000 rpm for 5 min for further LAMP tests. LAMP kits with primers, reagents, and reaction tubes (LMP207, Everplant Technology Ltd. Hong Kong, China) were used for LAMP assay. Next, 11.5 μL of deionized water and 3.5 μL of primer mix, as designed by Huang et al. [25] (Table 2), were added into the reaction tubes. Then, 10 μL of the supernatant was then extracted. Three replicates were prepared for each tree sample.

The prepared 25 μL samples were inverted 5 times and centrifuged at room temperature at 8000 rpm for 1 min to ensure all the chemicals were well mixed. The samples were then transferred into a PCR thermocycler at 65 °C for 1 h for DNA elongation and 85 °C for 5 min for DNA degradation. To visualize the results, the identification of fluorescence was operated in gel documentation (UVCI-1100, Major Science), which can demonstrate the brighter fluorescence. Turbid samples (precipitation reaction) with observable fluorescence were described as positive results, i.e., carried *P. noxius* (Figure 2A). On the other hand, clear samples (without precipitation reaction) with no fluorescence observed were defined as negative results, i.e., *P. noxius*-free (Figure 2B).

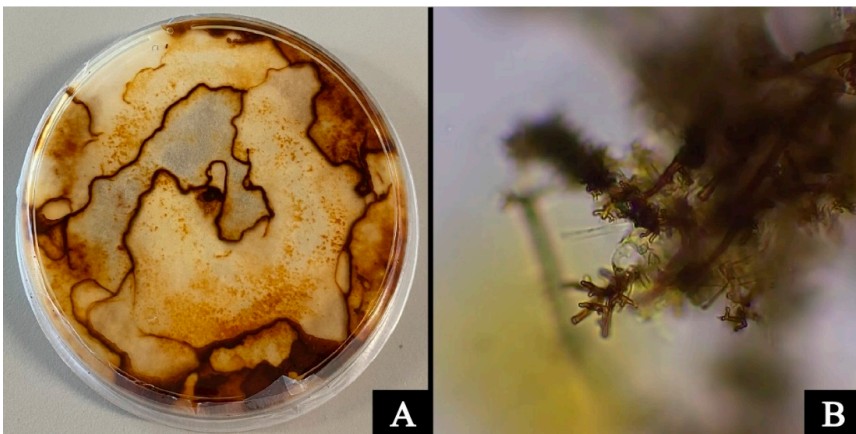

**Figure 1.** Visualization of colony of *Phellinus noxius* (Corner) G. H. Cunningham on PDA (**A**) by naked eye detection (brown color and irregular shape mycelium); and (**B**) under light microscope with 400× magnification (trichocysts produced).

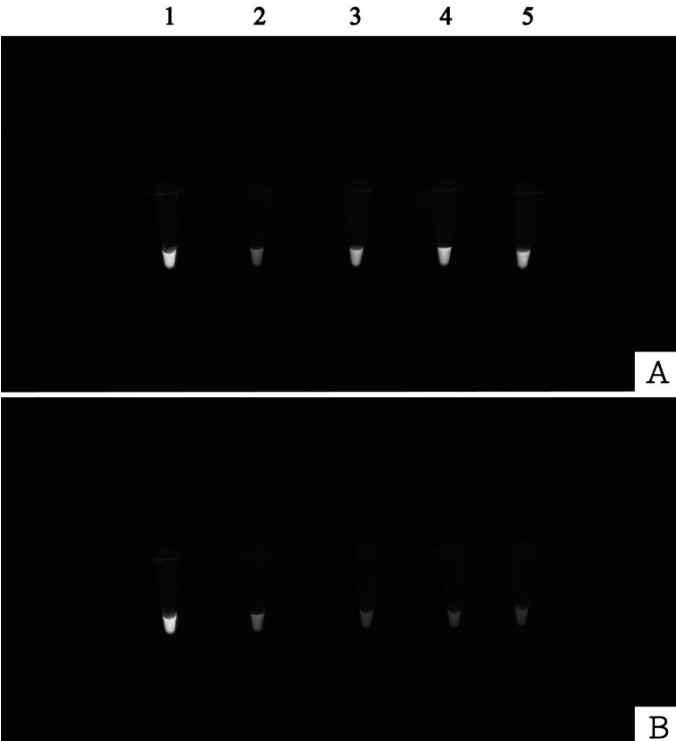

**Figure 2.** Visualization of (**A**) positive results and (**B**) negative results of *Phellinus noxius* (Corner) G. H. Cunningham in gel documentation using mLAMP and wLAMP. (**A**): Lane 1 indicated positive control, Lane 2 indicated negative control, Lanes 3–5 indicated positive results; (**B**): Lane 1 indicated positive control, Lane 2 indicated negative control, Lanes 3–5 indicated negative results.

**Table 2.** Five LAMP primers [1] used for detecting *Phellinus noxius* (Corner) G. H. Cunningham.

| Primers | Sequence (5′-3′) |
|---------|------------------|
| 1-F3 | TTTGAGGCCAAAGGTCAA |
| 1-B3 | GTGTCATGTTAATCTCAATACAACA |
| 1-LB | CAAGAGAAGCCGACTTACGC |
| 1-FIP | ACATTCACCGTTTACACTTGCTAATGTTAAGTGTTTGTCTCATTACAAGA |
| 1-BIP | TACACCAATTACTCGAGCAAAAGCTTAATATTGGACTTGGGGACTG |

[1] Reprinted/adapted with permission from Ref. [25]. 2014, Huang, Y.; Chen, C.; Hung, T.; Wu, M.L.; Chang, T.

### 2.3.3. LAMP Method Using Wood Chips (wLAMP)

PN3 samples inoculated for 4 to 5 days were used for the wLAMP. The other procedures were identical to the protocol of mLAMP as described in Section 2.3.2.

### 2.3.4. Statistical Analysis

All data analysis was conducted in Microsoft Excel 2016 for the analysis of the sensitivity and accuracy of LAMP in the diagnosis of BRRD by using mycelium and wood chips.

## 3. Results and Discussion

### 3.1. The Relationship between Sample Purity and LAMP Sensitivity

The results of fungal isolation were 100% matched with the provided diagnosis results of BRRD (Table 3). The results of mLAMP for BRRD-positive trees were 100% consistent with the confirmed results. In other words, no false negative results were observed by using mLAMP. Meanwhile, the method was considered as oversensitive as the agreement of results was reduced to 73.3% among BRRD-negative trees. For wLAMP, the results were 73.3% match for both BRRD-positive and -negative trees.

**Table 3.** Results of fungal isolation, mLAMP, and wLAMP of the 15 BRRD-positive trees and 15 BRRD-negative trees.

| Code [1] | BRRD Symptoms [1] | Fungal Isolation [2] | mLAMP [2] | wLAMP [2] |
|----------|-------------------|----------------------|-----------|-----------|
| | | BRRD-positive trees | | |
| P01 | Absence | + | + | − |
| P02 | Absence | + | + | + |
| P03 | Presence | + | + | + |
| P04 | Absence | + | + | + |
| P05 | Absence | + | + | + |
| P06 | Absence | + | + | + |
| P07 | Presence | + | + | + |
| P08 | Presence | + | + | + |
| P09 | Presence | + | + | − |
| P10 | Presence | + | + | + |
| P11 | Absence | + | + | − |
| P12 | Presence | + | + | + |
| P13 | Presence | + | + | + |
| P14 | Presence | + | + | + |
| P15 | Absence | + | + | − |
| | | BRRD-negative trees | | |
| N01 | Presence | − | − | − |
| N02 | Absence | − | − | + |
| N03 | Presence | − | + | − |
| N04 | Absence | − | − | + |
| N05 | Presence | − | − | − |
| N06 | Absence | − | + | − |
| N07 | Presence | − | − | − |

**Table 3.** *Cont.*

| Code [1] | BRRD Symptoms [1] | Fungal Isolation [2] | mLAMP [2] | wLAMP [2] |
|---|---|---|---|---|
| N08 | Presence | − | − | − |
| N09 | Absence | − | − | − |
| N10 | Absence | − | − | + |
| N11 | Presence | − | − | − |
| N12 | Presence | − | + | − |
| N13 | Absence | − | + | − |
| N14 | Absence | − | − | − |
| N15 | Presence | − | − | + |

[1] Refer to Table 1 for detailed information. [2] "+" indicated positive results; "−" indicated negative results.

High sample purity is pivotal to produce accurate results and is currently achieved by a series of rigid and time-consuming extraction processes [26,27]. Boesenberg-Smith et al. [28] reported that insufficient DNA purity could reduce the accuracy of PCR. Subramanian and Gomez [29] indicated the presence of undesirable substances could interfere with the normal DNA amplification in LAMP. Huang et al. [25] also observed that the low sample purity could restrict the sensitivity of LAMP. They therefore proposed the import of extracted DNA by purifying the DNA from 10 to 100 times to eliminate most of the undesired substances. However, this is time-consuming and technically demanding, and hence should be bypassed if possible. Our results for mLAMP echoes the findings of Almasi et al. [9]. They suggested that pure-cultured mycelium were well tolerant in LAMP and could be surrogates for stringent DNA extraction processes.

Compared to the mycelium, woods chips are more structurally complicated and mainly composed of secondary metabolites [30]. LAMP hitherto showed sufficient tolerance to direct blood [31], stool, urine [32], soil [33], and bacterial infected samples [34]. No study has applied wood chips directly to LAMP assay before. The direct use of wood chips offers an isolation-free method of diagnosis. In this study, the entire process of diagnosis only took 4–5 h, which is much shorter than the detection time in comparison to Huang et al. [25]. However, both false negative and false positive results were found in this study. The scarce amounts of DNA fragments in wood chips may impede the precise sample collection of *P. noxius*, resulting in false negative results. Meanwhile, there were two possible causes for the false positive results. First, magnesium ions in tree bark may promote the amplification of false DNA by exchanging with calcein-bonded magnesium ions, which exhibit bright green fluorescence under UV light [35,36]. Second, the undesired substances in the samples may induce a higher possibility of nonspecific amplification, which may occur through the primers and coincidentally bind to other non-target regions and amplify automatically [37].

### 3.2. The Effectiveness of LAMPS in Diagnosing BRRD

This study intended to evaluate the effectiveness of applying LAMPs for the early detection of BRRD. Fourteen trees without symptom expression were tested by LAMPs. Seven of the trees were known to be BRRD-positive and the others were BRRD-negative. Overall, the congruencies to fungal isolation results were 85.7% and 64.3% for mLAMP and wLAMP respectively among those symptomless trees. More specifically, mLAMP showed a 100% accuracy for identifying BRRD-positive trees, and 71.4% accuracy for BRRD-negative trees. For wLAMP, the accuracy was 87.5% for BRRD-positive trees, and dropped to only 57.1% for BRRD-negative trees. Both methods expressed high sensitivity in diagnosing BRRD-positive trees, be they symptomatic or without symptoms.

Early diagnosis of BRRD can minimize the dissemination of the disease, management cost, and risks to citizens. In recent years, different techniques and technologies have been developed to permit early diagnosis of BRRD. They include the detection of the odor of *P. noxius* by using trained dog [38] or electronic nose [39], and the use of rhizosphere microbiome as a biomarker to indicate the infection of *P. noxius* [40]. Yet the accuracy of these prompt methods with low operation threshold is highly subjective to the host species

and nearby abiotic factors, such as soil composition. The accuracy of LAMP, on the other hand, is independent of tree species and abiotic factors, allowing a prompt and sensitive diagnosis for BRRD.

### 3.3. The Incorporation of LAMPs in the Current Tree Management Work

Manpower shortage has long been an issue for urban tree management in Hong Kong. The current time-consuming diagnosis method of BRRD is not favorable for conducting large-scale screening and preventive diagnosis. The existing management strategy of BRRD should be retrofitted by introducing prompt diagnostic methods that requires low labor costs. It is demonstrated in this preliminary study that LAMP has a high tolerance to low sample purity and allows prompt diagnosis in less than 4 h, which is hours faster than the PCR method [25]. In addition, it allows an accessible threshold of operation compared with other molecular tests. The operators only have to be equipped with elemental techniques, such as disease identification, sample collection, and basic molecular biology skills. Because of the potential of LAMP, further studies can be conducted to further evaluate and improve the accuracy of both mLAMP and wLAMP methods.

According to our findings, it is anticipated that LAMP has a high potential to be incorporated into the current tree management in Hong Kong. The scheme of the incorporation could be divided into preparation, diagnosis, and control (Figure 3). The operators should prepare sterilized items such as handhelds, adze hand forks, and shovels before sampling. The details of the operation and precautions should be identical to Section 2.2.

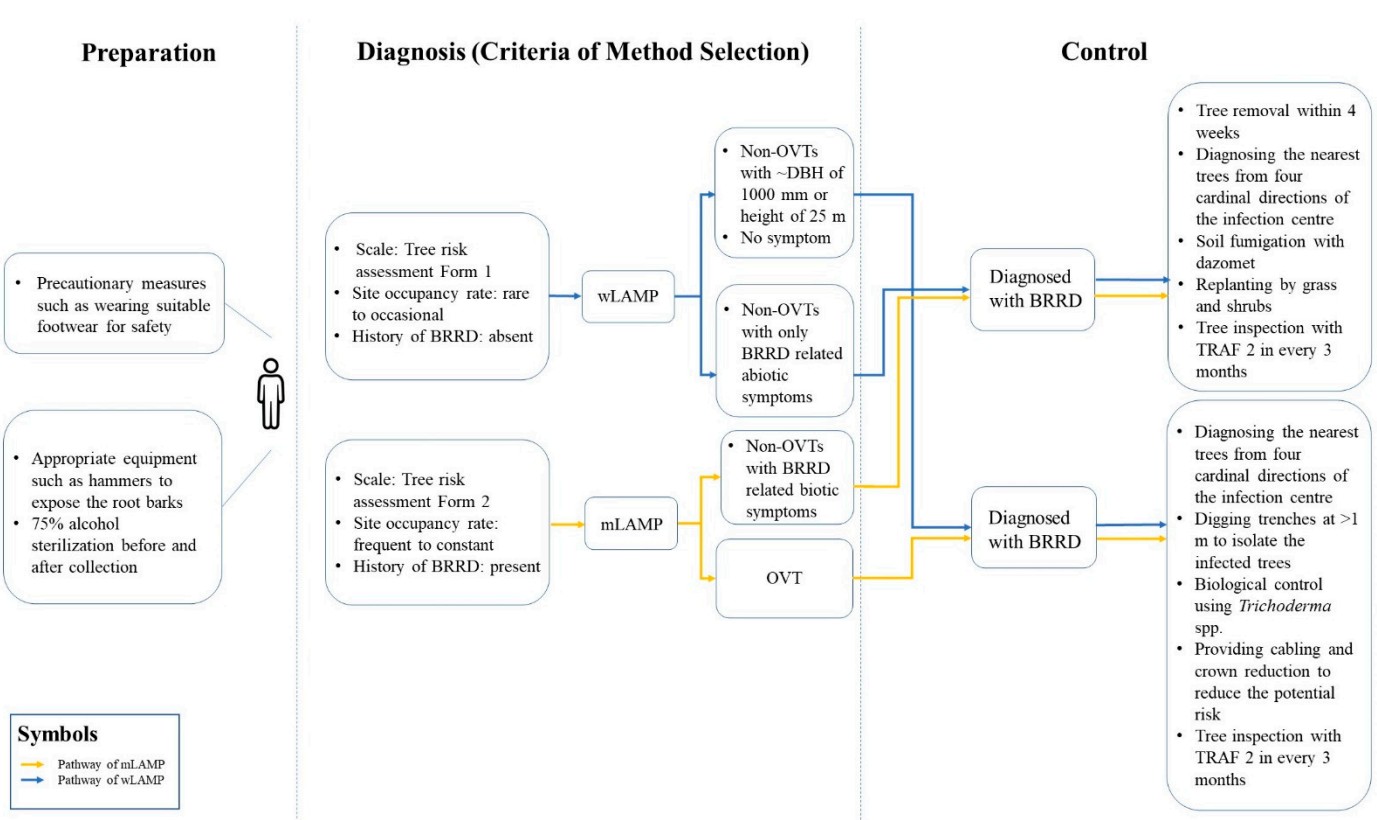

**Figure 3.** Incorporating LAMP as diagnostic tool in urban tree management in Hong Kong.

The selection of the BRRD diagnostic method relies upon the scale of diagnosis and site characteristics. wLAMP can be incorporated as an ancillary inspection item at the stage of Tree Risk Assessment Form (TRAF) 1 inspection in Hong Kong, which is a widely adopted local tree assessment methodology for broad scale monitoring of urban trees [41]. While wLAMP has a comparatively low sensitivity and specificity than mLAMP, it allows expeditious screening. Testing of 10–15 trees can be performed by an accomplished operator

per day. Furthermore, wLAMP can be applied to sites of lower risk, such as those with target areas that are with low traffic or pedestrian flow, and with the absence of BRRD history. We suggest applying wLAMP to trees without BRRD symptoms, but with approximate diameter at breast height (DBH) of 1000 mm or height of 25 m, which may pose considerable risk to human property or loss of ecological value [42]. Trees with abiotic symptoms of BRRD can be randomly sampled [2]. On the other hand, mLAMP encourages a broader sampling point and allows for more accurate detection. Therefore, its application should be limited to specific and severe cases. First, mLAMP can be used accompanied with TRAF 2 inspection in Hong Kong, which is applied for the individual trees for which detailed tree risk assessment are required [41]. Second, it should be applied to high-risk areas, that is, sites with target areas having high traffic and pedestrian flow or with presence of BRRD history. Third, mLAMP should be applied for trees with high significance, such as local registered Old and Valuable Trees [42]. The method should also be applied to trees with observable biotic symptoms of BRRD, such as mycelial nets and fruiting bodies.

Proper disease controls should be followed after the diagnosis of BRRD. Non-OVTs diagnosed with BRRD should be removed within four weeks after the inspection. *P. noxius* is reported to be able to survive in decayed root fragments for many years [2]. To terminate the dissemination of BRRD, the infected stumps and soils nearby should be excavated, sterilized, and removed. To eliminate BRRD, it is suggested to fumigate the soils with dazomet, which can kill *P. noxius* and pose minimum side effects to neighboring plants [43]. Simultaneously, since the pathogens are disseminated mainly through root-to-root contacts, the nearest trees from four cardinal directions of the infected trees should be monitored and diagnosed using mLAMP [44]. Immediate replanting is not recommended as *P. noxius* may not be removed exhaustively. Quarterly inspection of high-risk trees with TRAF 2 should be conducted to monitor the development of BRRD. For OVTs and other valuable trees, they should be isolated and controlled specifically. Similarly, mLAMP should be provided to inspect the nearest trees from four cardinal directions of the infected trees. Next, trenches should be dug at least one meter in depth to isolate the infected roots from neighboring healthy trees. Waterproof plastic clothes should also be installed in trenches. They restrict the development of infected roots and reduce the dissemination of disease [45]. However, isolation using trenches may only be suitable for gentle slopes. Furthermore, biological, and chemical controls are widely studied for their notable performance on BRRD control [46,47]. However, considering chemical controls such as Bordeaux mixture and propiconazole, which are detrimental to the environment, the adoption of *Trichoderma* spp. could be a surrogate to control the disease biologically [48]. Additionally, cabling and crown reduction should be performed to minimize the risk of the tree falling. According to the current guidelines, TRAF 2 inspection should be conducted every three months to assess the development of BRRD.

Trees tested with negative results are considered with the absence of BRRD. The presence of abiotic symptoms for these trees may be triggered by abiotic stress, such as nutrient deficiency, instead of BRRD infection.

## 4. Conclusions

LAMP is a sensitive and simple preliminary diagnostic tool that allows early and effective detection of BRRD. In this study, we evaluated the potential utilization of LAMP by using two types of sample materials (wood chips and mycelium) from 30 trees in 19 roadside slopes in Hong Kong. Through comparing the results with the isolation work, mLAMP showed 100% sensitivity for BRRD positive trees and 73.3% specificity for BRRD negative trees. At the same time, wLAMP yielded both false positive and false negative results. The result accuracy may be negatively affected by the undesirable substances and small sample size. The results of diagnosis on symptomless trees indicated the possibility of early diagnosis of BRRD. Based on the results of mLAMP and wLAMP in this study, the co-application of LAMP methods was elaborated in the current urban tree management work in Hong Kong. We envisage that the application of LAMP contains great potential for

current urban tree management in Hong Kong by accelerating the diagnosis process and facilitating large-scale screening of BRRD. Future studies should be conducted to enhance the specificity and sensitivity of the method. Meanwhile, a larger sampling group can be employed to evaluate the impacts of factors such as tree species and variety of bark types on the reliability of the method.

**Author Contributions:** Conceptualization, H.Z. and T.K.N.; methodology, T.K.N., K.C.L., Z.W.L., W.F.Y. and W.S.W.; software, Z.W.L.; validation, H.Z.; formal analysis, K.C.L., Z.W.L. and W.S.W.; investigation, K.C.L., Z.W.L. and W.F.Y.; resources, H.Z.; data curation, K.C.L., Z.W.L. and W.S.W.; writing—original draft preparation, T.K.N. and K.C.L.; writing—review and editing, H.Z.; visualization, K.C.L.; supervision, H.Z. and Z.W.L.; project administration, T.K.N. and W.F.Y.; funding acquisition, H.Z. and T.K.N. All authors have read and agreed to the published version of the manuscript.

**Funding:** This research was funded by Highways Department, HKSAR Government, Project No. THEi/CON-T/2019-07©.

**Institutional Review Board Statement:** Not applicable.

**Data Availability Statement:** Not applicable.

**Conflicts of Interest:** The authors declare no conflict of interest.

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
