# Peer review of "Development and Evaluation of Loop-Mediated Isothermal Amplification (LAMP) as a Preliminary Diagnostic Tool for Brown Root Rot Disease Caused by Phellinus noxius (Corner) G. H. Cunningham in Hong Kong Urban Tree Management"

_sustainability, doi:10.3390/su14159708_

Round 1

Reviewer 1 Report

this is an exciting study, significant disease, and updated technology.

Author Response

this is an exciting study, significant disease, and updated technology.

Response

Thank you for the positive comments.

Reviewer 2 Report

The manuscript is devoted to an actual problem - the diagnosis of brown
root rot of urban trees. The study used a new diagnostic method, namely
Loop Isothermal Amplification (LAMP) as a preliminary diagnostic tool.
The manuscript is well written and well structured. It is necessary to
correct or describe in detail the following points: - the table with the
results is moved to the Results and Discussion section, - the
statistical processing of the results is not presented, - the text in
Figure 3 is illegible, it should be enlarged. After making these
changes, the article can be accepted for publication in the journal
Sustainability.

Author Response

The manuscript is devoted to an actual problem - the diagnosis of brown root rot of urban trees. The study used a new diagnostic method, namely Loop Isothermal Amplification (LAMP) as a preliminary diagnostic tool. The manuscript is well written and well structured. It is necessary to correct or describe in detail the following points:

- the table with the results is moved to the Results and Discussion section

Response

Thank you for the suggestions. The table has been separated into two tables (Tables 1 and 2), with Table 1 showing the basic information of studied trees being kept in Materials and Methods section, and Table 2 showing the results being moved to the Results and Discussion section.

- the statistical processing of the results is not presented

Response

We conducted the statistical analysis in Microsoft Excel 2016. The information has been refined in lines 196-197.

- the text in Figure 3 is illegible, it should be enlarged

Response

We have enlarged the figure for suitable font size.

After making these changes, the article can be accepted for publication in the journal Sustainability.

Response

Thank you for the positive comments.

Reviewer 3 Report

Details about brown root rot disease (BRRD) management are inconspicuous. If the authors clear this information, it would be more interesting for the readers.

Abstract

L18-19- Year of study should be mentioned

L21- Study layout should be mentioned

L23 – Grammar mistake - For symptomless trees

Introduction

L33- BRR is caused by single pathogen species Phellinus noxius or there is existence of other pathogens as well?

L37 – Add recent reference

L45 - Mention few primary hosts of the disease

L61- Scientific proof is required

L61- Add a paragraph about losses that occur due to this disease in Hong Kong regions

L88- After diagnosis-how this disease is being controlled in the Hong Kong regions- discuss briefly

L93- how does LAMP tool of the study relate to the management of BRRD, are you doing control measures based on your results of study? Please make a coherent link between the two.

Materials and Methods

L98- When study was conducted? Please mention study year

L123- Which experimental design was adopted? Please also specify the number of replication

L132- For how many days inoculated samples were stored in an incubator?

 L133- 134- Morphological information of pathogen are not mentioned in the result section- please clear it

Results and discussion

L252- I think the main focus of study is Fig. 3. The elaboration of adopting LAMPs in the current urban tree management work in Hong Kong- This confusing frame of work does not support the result section. I recommended that Fig.3 in which some suggestions about disease management have been provided, must be included in the recommendation section in a good image form either in poor image form.   

 Discussion; add few recent references related to diagnosis of pathogen

All information about disease management should be included in a separate suggestion section-or disease management through antagonists should be incorporated in the materials and methods sections with details

Conclusion should be climax of all the headings mentioned in the Manuscript. It needs to be modified accordingly.

Author Response

Abstract

L18-19- Year of study should be mentioned

Response

We have refined the sentence to “In this study, 15 BRRD-positive and 15 BRRD-negative trees were sampled from 19 roadside slopes in Hong Kong during the end of 2020 to the mid of 2021” in lines 18-20.

L21- Study layout should be mentioned

Response

We have refined the sentence to “The wood tissues were isolated and cultivated in PN3 and PDA agars for the disease diagnosis. The mycelium samples in PDA were directly conducted in LAMP kits (mLAMP) to substitute the purified DNA materials. Wood tissues were also used in LAMP kits (wLAMP) as impurified and highly contaminated samples. The results of mLAMP and wLAMP were compared with the results of isolation to evaluate the specificity and sensitivity of LAMP method” in lines 20-23.

L23 – Grammar mistake - For symptomless trees

Response

We have revised the punctuation mark: “For symptomless trees.” -> “For symptomless trees,”.

Introduction

L33- BRR is caused by single pathogen species Phellinus noxius or there is existence of other pathogens as well?

Response

Yes, we limit the discussion to white rot fungi. Therefore, we have refined the sentences to “Phellinus noxius (Corner) G. H. Cunningham is a white rot fungus that can induce BRRD” in lines 37-38.

L37 – Add recent reference

Response

Two references have been added. They are:

Ann, P.; Chang, T.; Ko, W. Phellinus noxius Brown Root Rot of fruit and ornamental Trees in Taiwan. Plant Dis 2002, 86(8), 820-826.

Brooks, F. E. Brown root rot disease in American Samoa's tropical rain forests 2002. Pac Sci, 56(4), 377-387.

L45 - Mention few primary hosts of the disease

Response

Acacia confusa Merr., Ficus microcarpa L. f. and Prunus persica (L.) Batsch have been added as examples of host tree species in lines 50-51.

L61- Scientific proof is required

Response

Yes, we have done through the whole manuscript.

L61- Add a paragraph about losses that occur due to this disease in Hong Kong regions

L88- After diagnosis-how this disease is being controlled in the Hong Kong regions- discuss briefly

L93- how does LAMP tool of the study relate to the management of BRRD, are you doing control measures based on your results of study? Please make a coherent link between the two.

Response

We have refined the sentences to “In Hong Kong, the first case of BRRD was reported in 2008. In current years, the number of BRRD infected trees has increased exponentially. PCR and morphological diagnosis are the predominant methods but are limited from time and accuracy. The late and ineffective diagnosis could pose threat to human property due to tree falling. There is a need to establish a rapid and effective diagnosis method for the disease and incorporate it as a part of the tree risk management strategy. LAMP is a potential tool. Currently, most studies that applied LAMP are on human, animal, and crop pathogens. Studies focusing on the application of LAMP on urban trees for disease diagnosis are still scarce. Therefore, this study aimed to evaluate the accuracy and sensitivity of LAMP method in the diagnosis of BRRD by using two different types of sample materials, mycelium and wood chips, as well as explore the potential application of LAMP in urban forest management”.

Materials and Methods

L98- When study was conducted? Please mention study year

Response

We have refined the sentences to “Samples were collected from a total of 30 trees, with 15 trees with BRRD (BRRD-positive trees) and 15 trees free of BRRD (BRRD-negative trees), during the end of 2020 to the mid of 2021 (Table 1)”.

L123- Which experimental design was adopted? Please also specify the number of replication

Response

Random sampling of root and soil samples were adopted. There are 4 replications for each sample.

L132- For how many days inoculated samples were stored in an incubator?

Response

We have already mentioned the days of inoculated PN3 samples with 4 days and PDA samples with 5-6 days.

 L133- 134- Morphological information of pathogen are not mentioned in the result section- please clear it.

Response

The figure and morphological information are used for enhancing the accessibility of the method.

Results and discussion

L252- I think the main focus of study is Fig. 3. The elaboration of adopting LAMPs in the current urban tree management work in Hong Kong. This confusing frame of work does not support the result section. I recommended that Fig.3 in which some suggestions about disease management have been provided, must be included in the recommendation section in a good image form either in poor image form.  

Response

We have updated an image with higher resolution.

Discussion; 

Add few recent references related to diagnosis of pathogen

All information about disease management should be included in a separate suggestion section-or disease management through antagonists should be incorporated in the materials and methods sections with details

Response

Conclusion should be climax of all the headings mentioned in the Manuscript. It needs to be modified

We have updated the conclusion as follows: LAMP is a sensitive and simple preliminary diagnostic tool that allows early and effective detection of BRRD. In this study, we evaluated the potential utilization of LAMP by using two types of sample materials (wood chips and mycelium) from 30 trees in 19 roadside slopes in Hong Kong. Through comparing the results with the isolation work, mLAMP showed 100% sensitivity for BRRD positive trees and 73.3% specificity for BRRD negative trees. At the same time, wLAMP yielded both false-positive and false-negative results. The result accuracy may be negatively affected by the undesirable substances and small sample size. The results of diagnosis on symptomless trees indicated the possibility of early diagnosis of BRRD. Based on the results of mLAMP and wLAMP in this study, the co-application of LAMP methods was elaborated in the current urban tree management work in Hong Kong. We envisaged that the application of LAMP contains great potential for current urban tree management in Hong Kong by accelerating the diagnosis process and facilitating large-scale screening of BRRD. Future studies should be conducted to enhance the specificity and sensitivity of the method. Meanwhile, a larger sampling group can be employed to evaluate the impacts of factors such as tree species and variety of bark types on the reliability of the method.

Reviewer 4 Report

The authors report in this manuscript that they found an efficient diagnostic measures for the brown root rot disease by LAMP method. The information provided in the manuscript will be available worldwide and is worthwhile being published in the journal. However, some improvement should be required for the publication. I am listing comments on the manuscript individually below.

1. English

The authors should ask a native English speaker for the correction of English in their revised manuscript before resubmission.

L34

The abbreviation of the species does not need to be indicated. After spelled out completely, the species name can be referred to as just P. noxius.

L44

Give examples of the hosts.

L46

What does “infection zone” mean?

L98

Refer to the date of the study.

L108

Refer to the date of the collection.

Explain how these 30 trees were selected.

How were the BRRD infection in the 15 trees confirmed?

L122

Refer to the date of the experiments.

L132

Cite references for the growth.

L137-139

Who developed the identification procedure? Cite references, if the procedure followed any others reported elsewhere.

L148-151

How were the cubes collected and when?

Were the powder samples dissolved in the solvent together or individually?

This is the very point of this study, since the scientific evaluation of the research should be determined by the preparation of the samples. It should be kept in mind that any studies designed in inappropriate manners be never published.

L181-182

Even though positive results were obtained from BRRD-symptomless trees. the results could not be considered “false”, unless the pathogen had been confirmed in the trees. How can the results be concluded “false”?

L201

Can sampling error be considered for the false-negative results?

L225

What does “user-friendly” mean?

L232

Explain how time-consuming the method is.

L264

Spell out the words when appeared first, as “diameter at breast height (DBH)”.

 L277

Explain the “enormous tree form”, referring to actual sizes.

Author Response

The authors report in this manuscript that they found an efficient diagnostic measures for the brown root rot disease by LAMP method. The information provided in the manuscript will be available worldwide and is worthwhile being published in the journal. However, some improvement should be required for the publication. I am listing comments on the manuscript individually below.

  1. English

The authors should ask a native English speaker for the correction of English in their revised manuscript before resubmission.

Response

We have sought my institute Research Office to further improve the language. The supporting office is native English speaker from UK.

L34 The abbreviation of the species does not need to be indicated. After spelled out completely, the species name can be referred to as just P. noxius.

Response

Thank you for the comment. The indication has been removed.

L44 Give examples of the hosts.

Response

Acacia confusa Merr., Ficus microcarpa L. f. and Prunus persica (L.) Batsch have been added as examples of host tree species in lines 50-51.

L46What does “infection zone” mean?

Response

The sentence has been replaced by “The highly infectious nature of the disease can result in a widespread occurrence of the disease” for a clearer meaning.

L98 Refer to the date of the study.

Response

We have refined the sentences to “The study was carried out in Hong Kong, China (latitude 22°N and longitude 114°E) from 2020 to 2021”.

L108 Refer to the date of the collection. Explain how these 30 trees were selected. How were the BRRD infection in the 15 trees confirmed?

Response

We have refined the sentences to “Samples were collected from a total of 30 trees, with 15 trees with BRRD (BRRD-positive trees) and 15 trees free of BRRD (BRRD-negative trees), during the end of 2020 to the mid of 2021 (Table 1)”.

L122 Refer to the date of the experiments.

Response

We have refined the sentences to “The experiments were carried out in the laboratories of the Technological and Higher Education Institute of Hong Kong (THEi), Chai Wan, Hong Kong, during the end of 2020 to the mid of 2021”.

L132 Cite references for the growth.

Response

We have cited the reference from Westphal, K. R.; Heidelbach, S., Zeuner, E. J.; Riisgaard-Jensen, M., Nielsen, M. E.; Vestergaard, S. Z.; Bekker, N. S., Skovmark, J.; Olesen, C. K.; Thomsen, K. H.; Niebling, S. K.; Sørensen, J. L.; Sondergaard, T. E. The effects of different potato dextrose agar media on secondary metabolite production in Fusarium. International Journal of Food Microbiology 2021, 347, 109171.

L137-139 Who developed the identification procedure? Cite references, if the procedure followed any others reported elsewhere.

Response

As mentioned, we followed the identification procedure from Ann, P.; Chang, T.; Ko, W. Phellinus noxius Brown Root Rot of fruit and ornamental Trees in Taiwan. Plant Dis 2002, 86(8), 820-826.

L148-151 How were the cubes collected and when? Were the powder samples dissolved in the solvent together or individually? This is the very point of this study, since the scientific evaluation of the research should be determined by the preparation of the samples. It should be kept in mind that any studies designed in inappropriate manners be never published.

Response

We have updated the date of experiments to “during the end of 2020 to the mid of 2021”. The fungal isolation procedures have been updated as “To prepare samples for isolation, the wood tissue sample was first chopped into a few of 3 mm cubes and cleaned using deionized water to eliminate any undesired microorganisms. The samples were treated with sodium hypochlorite for 30 seconds for sterilization. The sodium hypochlorite was filtered, and the samples were further immersed in distilled water to remove attached sodium hypochlorite. The samples were collected and dried on a tissue paper. After drying, 4-6 pieces of wood chips were inoculated in PN3 agars (20 g malt-extract, 20 g nutrient agar, 30 mg benomyl, 10 mg dicloran, 100 mg ampicillin and 500 mg gallic acid per litre)”.

L181-182 Even though positive results were obtained from BRRD-symptomless trees. the results could not be considered “false”, unless the pathogen had been confirmed in the trees. How can the results be concluded “false”?

Response

The trees were diagnosed with predominant isolation method and the results were compared with the results from LAMP methods. We hypothesized fungal isolation is a reliable method with high accuracy.

L201 Can sampling error be considered for the false-negative results?

Response

There is the possibility.

L225 What does “user-friendly” mean?

Response

We have updated the sentence to “Yet, the accuracy of these prompt methods with low operation threshold is highly subjected to the host species and nearby abiotic factors”.

L232 Explain how time-consuming the method is.

Response

Such as DNA extraction and sequence method, etc.

L264 Spell out the words when appeared first, as “diameter at breast height (DBH)”.

Response

Thank you for the comment. The full spelling has been added.

L277 Explain the “enormous tree form”, referring to actual sizes.

Response

We have deleted the sentence including enormous tree form.

Round 2

Reviewer 3 Report

Paper may be accepted in present form